# Efficacy of Verbally Describing One’s Own Body Movement in Motor Skill Acquisition

**DOI:** 10.3390/brainsci9120356

**Published:** 2019-12-04

**Authors:** Tsubasa Kawasaki, Masashi Kono, Ryosuke Tozawa

**Affiliations:** 1Institute of Sports Medicine and Science, School of Human and Social Sciences, Tokyo International University, Kawagoe-City, Saitama 350-1198, Japan; 2Department of Rehabilitation, Murata Hospital, Ikuno-ku, Osaka-City, Osaka 544-0011, Japan; masashi1981@nifty.com; 3Department of Physical Therapy, Faculty of Health Science, Ryotokuji University, Urayasu-City, Chiba 279-8567, Japan; tozawa@ryotokuji-u.ac.jp

**Keywords:** verbally describing, motor imagery, motor skill acquisition

## Abstract

The present study examined whether (a) verbally describing one’s own body movement can be potentially effective for acquiring motor skills, and (b) if the effects are related to motor imagery. The participants in this study were 36 healthy young adults (21.2 ± 0.7 years), randomly assigned into two groups (describing and control). They performed a ball rotation activity, with the describing group being asked by the examiner to verbally describe their own ball rotation, while the control group was asked to read a magazine aloud. The participants’ ball rotation performances were measured before the intervention, then again immediately after, five minutes after, and one day after. In addition, participants’ motor imagery ability (mental chronometry) of their upper extremities was measured. The results showed that the number of successful ball rotations (motor smoothness) and the number of ball drops (motor error) significantly improved in the describing group. Moreover, improvement in motor skills had a significant correlation with motor imagery ability. This suggests that verbally describing an intervention is an effective tool for learning motor skills, and that motor imagery is a potential mechanism for such verbal descriptions.

## 1. Introduction

Acquiring motor skills is commonly achieved through the repeated execution of motor skills. However, recent research has indicated that the process of learning motor skills can be promoted through the use of cognitive strategies, without requiring actual physical movement [1,2]. The most common cognitive strategy used is motor imagery training, which is the mental practice of motor skills. The effectiveness of this strategy has been shown in research with individuals with a wide range of physical impairments, such as decreased fine motor skills [3], difficulties with balance and gait following a stroke [4], and complex regional pain syndrome [5,6,7]. Thus, clinical evidence for motor imagery training has been recently established. Motor imagery training has also been regarded as an effective approach for use in clinical settings, as it can be done without using actual physical movements.

The mechanisms of the beneficial effects of motor imagery training have been researched in the fields of both physiology and neurology. Particularly, previous research has demonstrated that the neural networks used during motor imagery are similar to those used during motor skill execution. These studies showed that when one imagines moving part of his or her body, the areas of the brain that correspond to that body part are significantly activated, including the primary motor cortex and dorsal premotor cortex, as well as hand representation located in the caudal cingulate motor area [8,9,10,11,12,13]. Additionally, excitability of the corticospinal system, as shown during transcranial magnetic stimulation, is increased during motor imagery [14,15,16,17]. Thus, neuroimaging and neurophysiological studies have supported the finding that motor imagery and motor execution involve overlapping neural structures in the central nervous system. Behavioral data have also shown the association between motor imagery and motor execution. For example, Decety and colleagues reported a high temporal correspondence between imagining walking and physically walking [18]. Bakker and colleagues, as an extension study of Decety’s report [18], demonstrated that the time used to imagine walking to a destination increased with increasing path length or decreasing path width [19]. These findings taken together with the previously mentioned research from multiple areas of study, show that motor imagery is related to motor execution. Thus, motor imagery training accesses similar mechanisms within the brain as motor skill execution, which facilitate motor prediction and simulation of motor performance. Based on the previous studies, motor imagery training is suggested to have beneficial results. 

In the present study, we focused on the learning of body movements through describing aloud one’s own motor performance to promote the acquisition of motor skills. Verbally describing one’s own body movements shows overlapping brain activities with motor execution and imagining one’s own body movements (motor imagery) [20]. Additionally, verbally describing one’s own motor performance includes motor imagery-related processes (recalling and analyzing one’s own motor skills) through inner speech and self-reflection [21,22]. The left frontal lobe of the brain, which is activated with inner speech [23] and self-reflection [24], is involved in providing motor imagery and motor programs [25]. Considering the previous research, we hypothesized that the process of verbally describing one’s own motor performance is related to motor imagery. In particular, we hypothesized that verbal descriptions are involved in motor imagery and as a result, motor learning may be promoted.

The aim of the present study was to examine whether verbally describing one’s own motor performance contributes to motor skill acquisition. To address this goal, the study was designed to demonstrate the efficacy of describing one’s own motor performance when compared to a control activity (reading a scientific magazine aloud). The second aim was to determine whether the describing intervention could be associated with motor imagery. Thus, we investigated the relationship between the efficacy of the describing intervention and motor imagery ability.

## 2. Materials and Methods

### 2.1. Participants

Participants included 36 healthy young adults (mean age = 21.2 years, SD = 0.7 years). All participants were strongly right-handed, as based on the Edinburgh handedness inventory [26]. Inclusion criteria were that participants had no previous experience with the ball rotation activity used in the intervention and no previous diagnosis of a neurological disorder. All participants gave informed consent prior to the study. The experimental protocols were approved by the Institutional Ethics Committee of Ryotokuji University (approval number 2622), and the tenets of the Declaration of Helsinki were followed.

### 2.2. Procedure

A schematic diagram of the procedure is shown in Figure 1. The participants sat comfortably in chairs, and they were asked to practice rotating two wooden balls clockwise in their left hand for five minutes. The balls were each 50 mm in diameter, weighed 37 grams, and had a relatively smooth surface. After practicing this action, the participants’ performance regarding the number of successful ball rotations and the number of ball drops in one minute was taken as a baseline measurement. They were instructed to rotate the two balls as quickly as possible during that minute. Next, the participants were assigned at random to one of two groups, the describing group (*n* = 18) or the control group (*n* = 18), and then informed of the details of the intervention. We ensured that there were no significant group differences in age, gender distribution, hand length (i.e., the length from the wrist to the top of the middle finger), Edinburgh handedness inventory score [26], or motor imagery ability (Table 1). Participants’ motor imagery abilities were assessed by examining mental chronometry [18] at a shoulder joint. This was done by measuring the time it took for both first-person perspectives imagining and executing shoulder flexion from a neutral position (i.e., their arms along the sides of their body) to 90 degrees, 10 times at free speed.

For the describing group, the participants were instructed by the examiner to describe aloud, in two minutes, how to perform the ball rotation task. The instructions were as follows: “Please verbally describe how to get successful ball rotations,” and, “What were the tips to get successful ball rotations?” The participants were also instructed to refrain from making any body movements. During the description, the examiner only provided back-channel feedback. Immediately after (Post 1), five minutes after (Post 2), and one day after (Post 3) the intervention, the participants’ ball rotation performances were measured. For the control group, the same procedure was followed as in the describing group; however, the participants in the control group read a scientific magazine aloud instead of participating in the verbally describing intervention, and they were not instructed in terms of verbally describing their own ball rotations.

In order to measure ball rotation performance, we timed their taped performance with a working stopwatch next to the participant’s hand. This allowed us to accurately count the number of successful ball rotations and ball drops by dividing the video into frames using video conversion software (Free Video to JPG Converter, DVDVideoSoft ltd., London, United Kingdom). The number of successful ball rotations and ball drops were calculated by dividing each second of the recordings into 30 frames. The recordings included a stopwatch in every frame. 

### 2.3. Dependent Variables and Data Analyses

The number of successful ball rotations and the number of ball drops were analyzed with two-way mixed analysis of variance with group (describing group, control group) as between-subject factor and session (baseline, immediately after, five minutes after, and one day after) as within-subject factor. A simple main effect test was applied if the interaction effect was significant. Bonferroni multiple comparison test was applied for post hoc comparison. In the describing group, the relationship between the improvement number of successful ball rotations in Post 3 (the number of successful ball rotations in Post 3 subtracted by the number of successful ball rotations in baseline) and the normalized values of the absolute error times of the mental chronometry (i.e., the relative error value) was investigated using Spearman’s rank correlation analysis. All the data were analyzed using SPSS 25. The level of significance was set at *p* < 0.05.

## 3. Results

Table 2 and Figure 2 show the results of the participants’ motor performance. For the number of successful ball rotations, the main effect of the session variable was significant (F (3, 102) = 33.59, *p* < 0.001, η^2^ = 0.50); however, there was no significant main effect of the group variable. A post hoc analysis of the session showed that the number of successful ball rotations in Post 1, Post 2, and Post 3 were significantly more than that in the baseline (Post 1 vs. baseline: *p* = 0.02, Post 2 vs. baseline: *p* < 0.001, Post 3 vs. baseline: *p* < 0.001). Additionally, the number of ball rotations in Post 2 and Post 3 were significantly more than that in Post 1 (Post 2 vs. Post 1: *p* < 0.001, Post 3 vs. Post 1: *p* < 0.001). A significant interaction (session × group) was obtained (F (3, 102) = 11.84, *p* < 0.001, η^2^ = 0.26). There were significant simple main effects of session variables in the describing group (F (3, 32) = 43.82, *p* < 0.001, η^2^ = 0.80) and the control group (F (3, 32) = 10.68, *p* < 0.001, η^2^ = 0.50). See Figure 2a for more details. In Post 3, the number of successful ball rotations in the describing group was more than in the control group (F (1, 34) = 7.57, *p* = 0.009, η^2^ = 0.18). Moreover, Spearman’s rank correlation analysis showed a significant relationship between the relative error value of the mental chronometry and the improvement number of successful ball rotations in Post 3 as shown in Figure 3 (*r* = −0.66, *p* = 0.003).

For the number of ball drops, no main effect for the variables of session and group was shown; however, there was a significant interaction (session × group) (F (3, 102) = 4.08, *p* = 0.009, η^2^ = 0.11). In the post hoc analysis for the describing groups, the number of ball drops in Post 3 was significantly lower than that in Post 1 (*p* = 0.02). Additionally, in Post 3, the number of ball drops in the describing group was significantly lower than in the control group as shown in Figure 2b (*p* = 0.005).

## 4. Discussion

The present study examined whether verbally describing own body movement demonstrates efficacy in motor skill acquisition when compared to reading aloud. The results showed that based on the number of successful ball rotations for one minute, positive effects were distinctly observed by the describing intervention. In the number of ball drops, the describing intervention led to a decrease only one day after the intervention. Rodgers et al. showed that both interventions using verbalization improved gross motor skill more than what was expected [27]. In the present study, these improvements in motor skill performance suggest that the describing intervention showed efficacy in fine motor skill acquisition. 

Improvements of each dependent variable indicate it was a meaningful change. We considered the number of successful ball rotations to reflect motor smoothness and the number of ball drops to reflect motor error. The describing intervention could show efficacy for both dependent variables, which is a trade-off between speed and accuracy. Our previous research examined the effects of intervention using motor imagery training combined with action observation by using the same paradigm as the present study [28]. The results of that study showed that the intervention had efficacy for motor smoothness, but not motor error. This combination intervention was assumed to enhance motor imagery, which involves the same mechanisms as the describing intervention. Considering the findings of the previous research, the impact of describing may be advantageous compared to interventions using motor imagery combined with action observation. However, in this regard, additional research should be conducted in the future.

A potential mechanism of the describing intervention could be attributed to motor imagery through recall and inner speech in terms of one’s own body movements. Recall and inner speech regarding one’s own ball rotation performance is required prior to verbally describing the experienced movement. Recalling one’s own body movement could directly enhance motor imagery. In addition, inner speech could help to analyze and clarify the described contents. Several previous studies have indicated that inner speech contributes to guiding behavior through the use of rules, procedures, and action steps [29,30]. Suwa reported that imagining and analyzing one’s own performance could be considered meta-cognitive verbalization, which can enhance motor skill acquisition [21,22]. The findings of the present study regarding the relationship between the relative error value of the mental chronometry and the improvement number of successful ball rotations supports the involvement of motor imagery in the efficacy of the describing intervention.

Our study had some strengths. First, usability of the describing intervention is higher than a conventional approach for some individuals. The describing intervention does not require strenuous physical activity, meaning it can be used with individuals who are not able to participate in high-intensity exercise therapy in clinical or sports settings such as frail older people or patients who have lower extremity orthopedic or cardiorespiratory problems in clinical and sports settings. Second, the describing intervention could have an advantage over other cognitive strategies. It has been pointed that certain populations (e.g., older people [31] or patients with brain injuries [32,33] or Parkinson’s disease [34,35]) have difficulty correctly imagining their own body movements. As opposed to interventions using motor imagery, the describing intervention is more accessible to these populations because the intervention has a distinct method. The impact of the describing intervention on motor skill acquisition may be higher than that of previously explored cognitive strategies such as action observation interventions.

Several limitations of our study should also be considered. First, we only investigated participants’ motor performance up to one day after the describing intervention. To determine essential motor skills acquisition, we should verify the effects for a longer period post-intervention, considering that motor learning is an internal process leading to relatively permanent changes. Second, it is possible that the effects of the describing intervention are related to task specificity. The ball rotation task has been frequently used in previous studies on motor learning [36,37,38,39,40,41]. However, the ball rotation task only tests finger movement coordination. Therefore, it is still unclear whether the describing intervention would still be effective when used with other kinds of motor skills (e.g., adaptive gait, finger tapping sequence learning task, or learning to throw objects). Therefore, further research into this issue is needed. Third, it is unclear whether the beneficial effects of describing differ depending on the level of expertise (novice vs. elite athletes). In this regard, more research should be conducted in the future. Finally, the process of describing body movement includes monitoring of the listener’s responses (including non-verbal presentation) and inferring mental and cognitive conditions. This process is likely related to theory of mind. Future research should investigate the different effects of verbal interventions such as self-instruction [42,43,44], which are not related to theory of mind.

In conclusion, the present study determined the beneficial effects of verbally describing one’s own body movement in motor skill acquisition. Additionally, the motor imagery would be related to the mechanism of the beneficial effects of the describing intervention. The describing intervention is expected to be applied in clinical and sports fields as an intervention that does not require strenuous physical activity.

## Figures and Tables

**Figure 1 brainsci-09-00356-f001:**
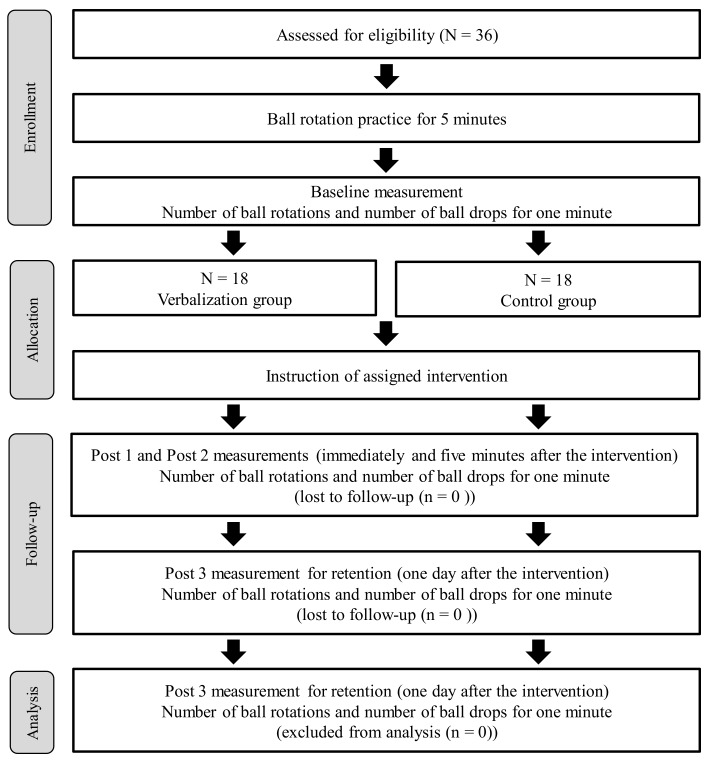
A schematic diagram of the procedure.

**Figure 2 brainsci-09-00356-f002:**
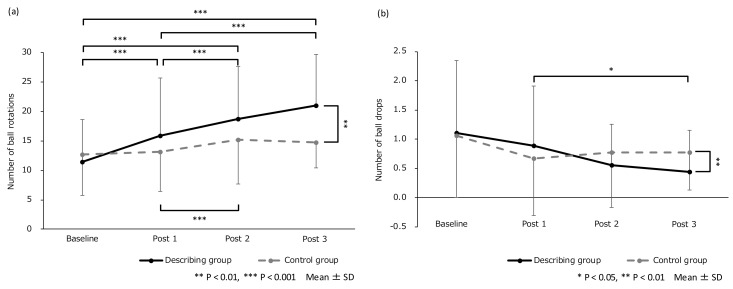
(**a**) Mean number of successful ball rotations in the describing and control groups. (**b**) Mean number of ball drops in the describing and control groups. For (**a**) and (**b**), the error bars denote the standard deviation.

**Figure 3 brainsci-09-00356-f003:**
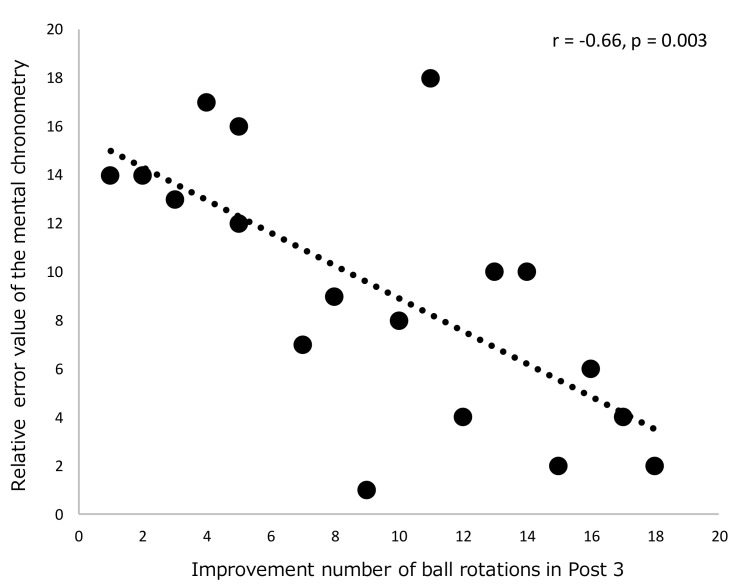
Scattergram showing the relationship between the improvement number of successful ball rotations in Post 3 and the relative error value of the mental chronometry in the describing group (*n* = 18, Spearman’s rank correlation).

**Table 1 brainsci-09-00356-t001:** Basic characteristics of each group.

	Describing Group (*N* = 18)	Control Group (*N* = 18)	*p*-Value
Age (years)	21.1 ± 0.83	21.4 ± 0.51	0.16
Gender distribution (% female)	33	33	1.00
Distance between wrist to top of middle finger at non-dominant hand (cm)	19.6 ± 2.8	19.3 ± 1.4	0.60
Edinburgh handedness inventory score	92.6 ± 6.6	89.9 ± 6.5	0.23
Absolute error of mental chronometry (s)	2.22 ± 1.23	1.37 ± 0.97	0.69

**Table 2 brainsci-09-00356-t002:** (a) Mean number of successful ball rotations (mean ± SD) and (b) mean number of ball drops (mean ± SD) for each group and session.

	Baseline	Post 1	Post 2	Post 3
(a)				
Describing group	11.46 ± 7.19	15.93 ± 9.77	18.71 ± 8.98	21.01 ± 8.72
Control group	12.67 ± 6.92	13.15 ± 6.73	15.24 ± 7.54	14.71 ± 4.27
(b)				
Describing group	1.11 ± 1.23	0.89 ± 1.02	0.56 ± 0.70	0.44 ± 0.70
Control group	1.06 ± 1.06	0.67 ± 0.97	0.78 ± 0.94	0.78 ± 0.65

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
