# Peer review of "Efficacy of Verbally Describing One’s Own Body Movement in Motor Skill Acquisition"

_brainsci, 2019, doi:10.3390/brainsci9120356_

Round 1
Reviewer 1 Report
The authors responded adequatly to all comments. I have no further remarks.
Author Response
Thank you reviewing our manuscript, brainsci-654160 "Efficacy of verbally describing one’s own body movement in motor skill acquisition" Also, we appreciate your giving us the opportunity for a revision. In an attempt to improve its structure and presentation, the manuscript has been revised based on your comments as well as those of the reviewers. In response to the following comments by reviewers. The revisions in the text are shown in red. The specific revisions made in response to comments from the reviewer are outlined below. We hope these revisions have made the manuscript suitable for publication in Brain Sciences.
We checked English spelling.
Reviewer 2 Report
This study analyzed the effects of verbally describing one’s own motor action on motor performances of 36 healthy young adults. The results showed improvements in motor smoothness when performing a ball rotation task in the describing subset group. Despite the paper is interesting and the message is clear, a major revision and some minor revisions should be considered to improve scientific quality.
Authors should consider a 2x4 two way mixed analysis of variance with group (describing and control) as between-subject factor and session as within-subject factor for the statistical analysis. This particular statistical analysis is suitable in comparing the mean differences between groups that have been split on independent variables over time. Moreover, the statistical analysis related to the simple main effects performed by authors in the significant interaction case should be reported and described in the materials and methods section. In fact, when the interaction effect results significant the mixed anova should be broken down in an unpaired t-test between the two groups and in a one way repeated measure analysis among sessions.
The Person’s correlation coefficient should be ranked according to: Dancey, C. P., and Reidy, J. (2007) Statistics without Maths for Psychology. London: Prentice Hall Paerson. A correlation coefficient r of 0.53, line 144, indicates a moderate positive correlation between the normalized value of the absolute error time of the mental chronometry and the improvement number of ball rotations in post 3. However, in figure 3 a negative correlation coefficient is reported.
Moreover, the term “the normalized values of the absolute error time of the mental chronometry” could be confounding. I suggest replacing it with the term relative error.
Line 31-32 “However, recent research... actual physical movement”. This sentence should be supported by a reference.
Table 1 “Absolute error of mental chronometry (sec)”. Please replace sec with s as required by the International System Unit recommendations.
Line 175- 177 “Our previous research... as the present study”. The reference 25 is not related to authors’ previous research but it is referred to Wrobel et al. Please check.
Author Response
Thank you reviewing our manuscript, brainsci-654160 "Efficacy of verbally describing one’s own body movement in motor skill acquisition" Also, we appreciate your giving us the opportunity for a revision. In an attempt to improve its structure and presentation, the manuscript has been revised based on your comments as well as those of the reviewers. In response to the following comments by reviewers. The revisions in the text are shown in red. The specific revisions made in response to comments from the reviewers are outlined below. We hope these revisions have made the manuscript suitable for publication in Brain Sciences.
Authors should consider a 2x4 two way mixed analysis of variance with group (describing and control) as between-subject factor and session as within-subject factor for the statistical analysis. This particular statistical analysis is suitable in comparing the mean differences between groups that have been split on independent variables over time. Moreover, the statistical analysis related to the simple main effects performed by authors in the significant interaction case should be reported and described in the materials and methods section. In fact, when the interaction effect results significant the mixed anova should be broken down in an unpaired t-test between the two groups and in a one way repeated measure analysis among sessions.
Response: We performed two-way mixed ANOVA (group×session) in resubmission 1. We revised manuscript to clarify this point including post hoc analysis (L122).
The Person’s correlation coefficient should be ranked according to: Dancey, C. P., and Reidy, J. (2007) Statistics without Maths for Psychology. London: Prentice Hall Paerson.
Response: Thank you for suggestion. Spearman’s rank correlation analysis was conducted based on the comment (L130), as a result, correlation coefficient was higher than previous result (L146, figure 3).
A correlation coefficient r of 0.53, line 144, indicates a moderate positive correlation between the normalized value of the absolute error time of the mental chronometry and the improvement number of ball rotations in post 3. However, in figure 3 a negative correlation coefficient is reported.
Response: Actually, the correlation analysis showed negative correlation. Thus, the correlation coefficient r is -0.53. We revised the value in manuscript (L146, figure 3).
Moreover, the term “the normalized values of the absolute error time of the mental chronometry” could be confounding. I suggest replacing it with the term relative error.
Response: Thank you for your suggestion. We revised this point (L129, Figure 3).
Line 31-32 “However, recent research... actual physical movement”. This sentence should be supported by a reference.
Response: This sentence was supported by references (L31).
Table 1 “Absolute error of mental chronometry (sec)”. Please replace sec with s as required by the International System Unit recommendations.
Response: Thank you for pointing this out. We revised this point (Table 1).
Line 175- 177 “Our previous research... as the present study”. The reference 25 is not related to authors’ previous research but it is referred to Wrobel et al. Please check.
Response: Thank you for pointing this out. We revised this point (L179).
Round 2
Reviewer 2 Report
All the comments have been solved by the authors